# The Impact of Radiotherapy and Attenuated Chemotherapy Regimens in Older Patients with Classic Hodgkin Lymphoma: A Real-Life Study from the ReLLi Network

**DOI:** 10.3390/cancers17050765

**Published:** 2025-02-24

**Authors:** Maria Christina Cox, Matteo Caridi, Alexandro Patirelis, Ilaria Del Giudice, Alessandro Pulsoni, Daniela Renzi, Sabrina Pelliccia, Roberta Battistini, Paola Anticoli Borza, Ombretta Annibali, Vito Rapisarda, Eleonora Alma, Nadia Messina, Gianna Maria D’Elia, Francesco Marchesi, Natalia Cenfra, Maria Paola Bianchi, Fiammetta Natalino, Andrea Carpaneto, Giovanni Manfredi Assanto, Anna Giulia Zizzari, Elena Maiolo, Vitaliana De Sanctis, Stefan Hohaus, Luigi Rigacci

**Affiliations:** 1Hematology, Department of Hematology and Oncology, Tor Vergata University Hospital, 00133 Rome, Italy; 2Division of Hematology, Belcolle Hospital, 01100 Viterbo, Italy; 3Institute of Hematology and Center for Hemato-Oncology Research, University of Perugia and Santa Maria Della Misericordia Hospital, 06129 Perugia, Italy; 4Thoracic Surgery, Department of Surgical Sciences, University of Rome Tor Vergata, 00133 Rome, Italy; alexandro.patirelis@hotmail.it; 5Hematology, Department of Translational and Precision Medicine, Sapienza University of Rome, 00185 Rome, Italy; 6UOC Ematologia, AOU Policlinico Umberto I, 00161 Rome, Italy; 7UOC Ematologia, Ospedale Santa Maria Goretti, 04100 Latina, Italy; 8Hematology and Stem Cell Transplant Unit, IRCCS Regina Elena National Cancer Institute, 00144 Rome, Italy; 9Hematology, Azienda Ospedaliera Universitaria Sant’Andrea, 00189 Rome, Italy; 10Hematology, Azienda Ospedaliera San Camillo, 00152 Rome, Italy; 11Azienda Ospedaliera San Giovanni Rome, 00184 Rome, Italy; 12Hematology, Campus Biomedico, 00128 Rome, Italy; 13Hematology, IRCCS Policlinico Gemelli, Università Cattolica del Sacro Cuore, 00168 Rome, Italyelenam86@hotmail.it (E.M.); 14Radiotherapy, Azienda Ospedaliera Universitaria Sant’Andrea, 00189 Rome, Italy

**Keywords:** classic Hodgkin lymphoma, elderly, chemotherapy, radiotherapy, synthetic data, survival, toxicity, anthracyclin, ICI, Brentuximab

## Abstract

Subjects aged ≥60 years with classic Hodgkin lymphoma (eHL) represent a distinct subgroup of Hodgkin lymphoma, differing clinically, biologically, and prognostically. In fact, eHL has historically been associated with poorer outcomes compared to younger patients. This study explores the outcomes of 150 eHL patients enrolled in a multicenter, real-life dataset that includes a total of 751 patients diagnosed between 2013 and 2018. Both traditional statistical analyses and machine learning algorithms were used to generate synthetic data. We highlight that while patients aged 60–69 treated with curative intent achieve excellent long-term outcomes, those aged ≥70 remain an unmet clinical need, experiencing worse outcomes even after achieving complete remission (CR). Another key finding of this study is the positive impact of radiotherapy on the overall survival of older patients, as well as the comparable efficacy of reduced-dose versus standard-dose anthracycline-containing regimens. The introduction of novel antibodies, particularly in sequential regimens, is expected to improve outcomes in eHL.

## 1. Introduction

Patients aged ≥60 years with classic Hodgkin lymphoma (cHL), referred to as elderly Hodgkin lymphoma (eHL), account for approximately 20–25% of all cHL cases. The clinical and biological characteristics of eHL subjects distinguish them from younger patients, contributing to historically poorer outcomes. Notably, 5-year progression-free survival (PFS) and overall survival (OS) rates are consistently lower in eHL compared to those of patients under 60.

The development of novel, potentially less toxic therapies has renewed interest in this patient population. A deeper understanding of contemporary treatments and outcomes is essential for planning new combination therapies and benchmarking real-world results against emerging drugs. Most data on eHL originate from Northern Europe [1,2,3] and North America [4,5], with a notable paucity of studies from Southern Europe. Furthermore, GLOBOCAN data (https://gco.iarc.fr/today, accessed on 1 September 2024) indicate that Hodgkin lymphoma mortality rates in Italy surpass those of many other Western European countries.

The largest published series on eHL primarily focus on patients diagnosed since 2000 [1,2,3,5]. However, from 2000 to 2010, toxicity-reduction measures such as omitting Bleomycin from ABVD following interim PET scans, extensive use of growth factors, and involved field radiotherapy were not widely implemented [4,5]. To address this gap, we analyzed the epidemiology, chemotherapy treatment, and outcome of a consecutive series of eHL patients diagnosed between 1 January 2013 and 31 December 2018, when Brentuximab-vedotin (BV) and checkpoint inhibitors (CPI) were not yet available.

Data were retrieved from the registry of the Rete Linfomi Lazio (ReLLi), providing valuable insights into this underrepresented population.

## 2. Materials and Methods

This is a retrospective analysis using the Lazio Region Network (ReLLi) database, which relies on 13 academic and non-academic institutions in this region of Italy. Patients with HL older than 18 years of age consecutively treated between 2013 and 2018 were included. Disease characteristics and demographic data (i.e., Ann Arbor staging, EORTC classification of limited vs. advanced stage, extranodal disease, and bone marrow involvement), laboratory values, first- and second-line chemotherapy, type of responses, date of progression, and date of death or time of last follow-up were collected from medical records and entered into the ReLLi centralized database. We considered bulky lesions with a diameter greater than 7.5 cm. The efficacy of treatments was assessed according to the international working group criteria of the National Cancer Institute, established in 1999 [6], and, more recently, according to the Lugano Classification [7]. This study has been approved by the ethical committees of the participating institutions and is in agreement with the Declaration of Helsinki. This study is based on patients’ follow-up through to 1st December 2023. The analysis of epidemiological data was carried out on the overall population of 751 subjects, while detailed survival analyses were limited to patients ≥60 years. Primary endpoints were overall (OS), event-free (EFS) progression-free (PFS), disease-free (DFS), and cancer-specific survival (CSS). Secondary endpoints were the type of first-line treatment, adherence to the treatment plan, and response to treatment. PFS was defined from the date of 1st-line treatment to the date of relapse, progression, death from any cause, or date of the last contact. OS and CSS were calculated from the date of first-line therapy to the date of the last follow-up or death from any cause or lymphoma, and treatment toxicity, respectively. EFS was defined from the date of first-line treatment to the date of relapse, progression, death, failure of first-line treatment, treatment discontinuation from any cause, or date of the last contact. DFS was defined from the date of complete remission (CR) following first-line treatment to the date of relapse, progression, death, or date of the last contact. Data were analyzed with descriptive statistics. For numeric variables, medians and ranges are reported. For categorical variables, the number and percentages of patients in each category are reported. The Kaplan–Meier method was used to plot survival curves. The influence of individual risk factors on EFS, PFS, OS, and CSS was tested with log-rank tests and described with hazard ratios (HRs) from Cox proportional hazard models. Multivariable models including all risk factors were also fitted. Age was categorized into two groups in the univariable models to match the graphical. Separate Kaplan–Meier curves were drawn by age group, disease stage, consolidative radiotherapy, and reduced-dose anthracycline schedules vs. full-dose ones. Data were analyzed using the statistical software SPSS vs 6 (IBM SPSS Statistics, Chicago, IL, USA).

Differences were considered statistically significant when the *p*-value was 0.05 or less (two-sided test). Synthetic data is artificial data generated by machine learning algorithms trained to learn the essential characteristics of a real source dataset, allowing an increase in information by data augmentation and integration [8,9,10]. Synthetic data were generated with R Studio vs 4.3 3 (R Studio: Integrated Development for R. R Studio, PBC, Boston, MA, USA) using the “synthpop” package [11]. Statistical analyses were performed through the packages “survival” [12,13] and “survminer” [14]. Graphics were generated with ggplot2 [15,16]. Considering the size of the dataset, synthesis with classification and regression trees rather than random forest algorithms was chosen. The syn.ctree and syn.cart functions were applied. They differ, among other factors, in the selection of a splitting variable and a stopping rule for the splitting process [11]. Two datasets were thus generated using these two approaches. The dataset containing the lowest pMSE value and the S_pMSE value closest to 1 was analyzed. For better data generation, it is suggested to have a sample size equal to “100 + 10 × no. of variables used in modeling the data”. We preferred to focus on eHL; therefore, our sample size was 150. We needed to generate synthetic data using a maximum of 5 variables. We aimed to create several sub-datasets with the main information needed for both robust data synthesis and the generation of synthetic datasets adequate for our investigations. Due to the great importance of age in our dataset and its clinical link with many other variables, age was included as a variable in every dataset chosen for data augmentation. Time to event (death, progression, etc.) and the event status (0–1) were included in each sub-dataset. In the chemotherapy exploratory dataset, the other variables included were “first-line therapy”, “ABVD-like regimens” (i.e., full-dose anthracycline-based chemotherapy), “attenuated regimens” (schedules that alternate cycles of full-dose anthracycline with cycles that do not include anthracyclines, such as OPP and COPP. In the radiotherapy exploratory dataset, the other variables were “first-line radiotherapy” (0–1) and “disease stage” (early or advanced). In order not to add another variable, we generated synthetic data considering only patients treated with curative intent.

## 3. Results

### 3.1. Patient Features

Overall, 751 cHL patients were enrolled, and 150 patients (21%) were aged ≥60 years. Of these, 100 (67%) were treated at academic hospitals. The median age of eHL subjects was 70.5 years (range: 60–89.8 years). Patients aged ≥70 represented 52.6% of this subset. PET-CT staging was conducted in 106 patients (70.6%), while the remainder underwent contrast-enhanced CT scans.

Among eHL patients, 55 (36.6%) had early-stage disease, but only three (2%) were in stage I without bulky disease. A significant proportion (25%) presented with more than three nodal sites. Bulky disease (≥7.5 cm) was observed in 5% of cases. Compared to patients under 60, eHL patients had higher rates of advanced-stage disease (OR 2.148; *p* < 0.0001), mixed cellularity (OR 2.8; *p* < 0.0001), and lymphocyte-depleted classic HL (OR 5.502; *p* = 0.0006), and lower frequencies of nodular sclerosis (OR 0.3036; *p* < 0.0001). eHL patients exhibited a lower prevalence of bulky disease (OR 2.346; *p* < 0.0004) and B symptoms (*p* = 0.1402). A male predominance was observed in eHL (OR 0.6124; *p* = 0.0082), alongside increased rates of hypoalbuminemia (<3.5 mg/dL) and anemia (Hb < 10.5 g/dL) (Table 1).

### 3.2. Treatment Approaches and Outcomes

#### 3.2.1. Early-Stage Treatment

Of the 55 early-stage patients, 37 (67.2%) received combined chemotherapy and radiotherapy, 17 (31%) underwent chemotherapy alone, and 1 patient received radiotherapy only. Among those receiving combination therapy, 29 (78%) achieved complete remission (CR), 6 (16%) partial remission (PR), and 2 (6%) showed no response.

Fifteen patients treated with chemotherapy alone achieved CR, with ten receiving two additional cycles of systemic therapy. Relapse rates were higher in the chemotherapy-only group (46.6%) compared to those receiving combination therapy (14.2%).

#### 3.2.2. Advanced Stage Treatment

Among 95 advanced-stage patients, 77 (81%) were treated with curative intent. Fifty-one received ABVD/AVD, eight underwent MBVD/MVD, and ten received attenuated regimens. Fifteen patients received palliative therapy. Fifty-five patients (71.4%) achieved CR following curative-intent treatment, five (6.4%) PR, and thirteen (16.8%) showed no response. Three patients died during treatment.

Of the 59 patients receiving full-dose anthracyclines, 45 (76.2%) completed six cycles. Among those treated with palliative intent, CR and PR rates were 18.7% each, while 62.6% experienced progression. First-line chemotherapies are summarized in Table 2.

### 3.3. Second-Line Therapies

Seventy-two patients (48%) experienced relapse or were refractory after a median follow-up of 85 months. Data on second-line therapy (Table 3) were available for 32 patients (44%). Of these, 22 (68.7%) received active treatment, and 9 underwent autologous stem cell transplantation (auto-BMT) after achieving CR. Ten patients received palliative care. Following salvage treatment, 10 patients (31.2%) achieved CR and 5 (15.6%) PR, while 17 (53.1%) did not respond.

### 3.4. Outcomes and Survival Analysis

Following a median observation time of 81 months (range = 60–129), sixty (40%) patients died, and in 90% of them, death was lymphoma-related. The incidence of early deaths due to treatment toxicity was 3.3%. The 5-year overall survival (OS) was 87% for patients aged 60–69 and 62% for those aged ≥70 (Appendix A). Among 132 patients (88%) treated with curative intent, the 5-year cancer-specific survival (CSS) was 93% for the 60–69 group and 70% for the ≥70 group, while event-free survival (EFS) was 78% and 58%, respectively (*p* < 0.001). All these differences were statistically significant (i.e., <0.001, Figure 1).

The following factors were analyzed for outcome: age 60–69 vs. ≥70 y; Gender; stage I–II vs. III–IV, Bulky ≥ 7.5, B-symptoms, HB < 10.5 g/dL, WBC ≥ 15,000 mcL; Radiotherapy yes vs. no, CR after 1st line therapy, interim PET positivity (Table 4).

Advanced stage was a significant prognostic factor for OS in the ≥70 years (*p* < 0.001) but not the 60–69 years subset (*p* = 0.39, Appendix A). The omission of radiotherapy in patients treated with curative intent (Appendix A) was a negative factor for OS, in both the early- (*p* = 0.007) and the advanced-stage group (*p* = 0.048) (Table 4 and Appendix A). The event-free survival (EFS) of patients who were treated with full-dose anthracycline was not different from the subset who received an attenuated approach (Appendix A). However, in subset analysis, a difference was observed (*p* = 0.045) in the ≥70 years subgroup (Appendix A). In multivariate (MV) analysis, significant factors for EFS, OS, and CSS were age ≥70, failure to achieve CR, and the omission of consolidative radiotherapy (Table 4). However, in MV analysis, between the 60–69 years and the ≥70 years subsets, the omission of radiotherapy was significant only for the older age group (Appendix A).

### 3.5. Synthetic Data Augmentation

To address subgroup size limitations, synthetic data were generated using the “synthpop” package in R Studio. A total of 450 synthetic patients were generated (300% data augmentation) to strengthen analyses. The median OS in the synthetic cohort was 102 months for the ≥70 group and 129 months for the 60–69 group, mirroring real-world results. Synthetic data confirmed the non-inferiority of attenuated chemotherapy (*p* = 0.095) and the negative impact of omitting radiotherapy (*p* < 0.001) (Figure 2a,b and Figure 3). This approach validated the robustness of findings and reduced potential bias.

## 4. Discussion

To our knowledge, this is the first real-life survey of elderly Hodgkin lymphoma (eHL) patients from Southern Europe, treated with regimens that could still be considered standard. However, our data pertain to an era when BV and CPI were not yet indicated for HL, as ongoing trials were still investigating the efficacy of new combinations.

Overall, this study shows that treatment with ABVD-like schedules results in satisfactory outcomes for patients aged 60–69 but not for those aged 70 and older with advanced-stage disease. Furthermore, we observed that the omission of radiotherapy was one of the most significant factors contributing to poor event-free survival (EFS) and cancer-specific survival (CSS), particularly in localized stages. This confirms the critical role of radiotherapy in the therapeutic management and outcomes of eHL patients. To our knowledge, the impact of this factor on survival of eHL patients has not been analyzed in other real-life series [2,4].

The percentage of early-stage patients who received radiotherapy (~67%) is similar to [4] or slightly higher [2] than those reported by Orellana-Noia et al. and Moccia et al. Presumably, RT was administered only to patients with residual lymphadenopathy.

In multivariate (MV) analyses (Table 2), the only significant factors for EFS, CSS, and overall survival (OS) were age ≥ 70, failure to achieve complete remission (CR), and not receiving radiotherapy. In a subset analysis, the omission of radiotherapy remained highly significant only in the ≥70 age group Appendix A. These data suggest that RT, in subjects aged ≥70, with limited stage, may be beneficial even when there are no residual lesions and interim PET is negative. Only 7% of ≥70-year-old patients in advanced stage received RT; this is presumably due to the very low incidence of bulky disease in eHL compared to younger patients (Table 1).

Our database was reanalyzed through the creation of a synthetic dataset, which confirmed the impact of radiotherapy in eHL patients at both early and advanced stages. In fact, performing valid bioinformatic analyses requires a significant amount of real-world data. Indeed, the creation of synthetic data may reduce the risk of bias from the limited number of subjects analyzed. The synthetic cohort confirmed and strengthened the impact of RT on OS.

This further highlights the critical role of radiotherapy in the therapeutic management and outcomes of eHL patients. Three randomized trials (the H10 trial of the EORTC, the HD16 trial of the GSHG, and the RAPID trial from the UK) demonstrated that the omission of radiotherapy in patients with limited-stage disease and an early complete metabolic response is associated with inferior progression-free survival [17,18,19,20], but failed to show an impact on OS. These trials also included a minority of elderly patients: up to 70 years in the H10 trial and 75 years in the HD16 and RAPID trials. A detailed analysis of this small subgroup of elderly subjects has not been reported. As far as we know, the impact of radiotherapy on OS has not been analyzed in other real-life series focused on eHL [2,4]. Therefore, we believe that radiotherapy is unjustifiably underutilized in elderly patients [2,4], likely due to (1) concerns about poor tolerance and acute adverse events associated with radiotherapy, and (2) logistical barriers.

Full-dose ABVD-like schedules were administered to 77.4% of the 60–69 age subgroup and 73.3% of those aged ≥70 (Table 3). Notably, bleomycin reduction after two ABVD cycles was performed in only 8 (13.5%) of 59 advanced-stage patients, likely because data from the RATHL study were not widely available during patient enrollment. Johnson et al. [21] demonstrated that, with a PET-oriented approach, it is possible to remove bleomycin (AVD) after two cycles of ABVD without reducing treatment efficacy in terms of both PFS and OS [21]. Moreover, respiratory adverse events were more severe in the ABVD arm than in the AVD arm. The German Hodgkin Study Group (GHSG) also showed that, in localized stages, limiting bleomycin—particularly in elderly patients—to the first two cycles of ABVD was less toxic while maintaining comparable efficacy. Recently, the German group eliminated bleomycin, among other agents, in the new BrECADD scheme [22,23].

An attenuated regimen with a 50% reduction in anthracycline was administered to 14% of subjects (Table 3). This was not associated with worse PFS (Appendix A). The synthetic dataset analysis confirmed these results, supporting the notion that less intensive chemotherapy may effectively control disease and improve long-term survival in older and frail patients. Additionally, patients who achieved remission following a palliative approach experienced prolonged OS. Nevertheless, chemo-refractoriness remains a significant issue in eHL. Indeed, we believe that only newer regimens, incorporating targeted drugs, will improve the outcome of refractory patients.

Recently, the results of several trials enrolling eHL and based on new combination regimens have become available [24,25]. Indeed, the introduction of new antibodies, particularly in sequential regimens, is expected to improve outcomes in this subset [24]. In fact, it was recently reported that AAVD is less effective and more toxic in older populations [26]. Conversely, Evens’ study [27] demonstrated that the sequential use of immunotherapy with chemotherapy enhances results in elderly patients by reducing toxicity and enabling more patients to complete treatment without adverse events leading to discontinuation. Also, the results of the pivotal phase III SWOG S1826 study [28] were published. Although this trial was not specifically designed to address the treatment of elderly patients, a sub-analysis clearly demonstrated that in fit older patients (over 60 years), the N-AVD combination resulted in significantly better outcomes compared to BV-AVD, both in terms of tolerability and efficacy [28]. However, Nivo-AVD may increase the risk of immune-related cardiotoxicity [28,29].

Currently, beyond BV and CPI, a broader use of systematic geriatric assessments to identify patients capable of tolerating full-dose therapies, along with an expanded use of radiotherapy, could improve success rates in eHL [30].

Few data are available regarding laboratory parameters in older eHL and their impact on prognosis [31]. Both anemia and a low lymphocyte count are included in the IPS score [32], which, after the advent of the interim-PET Deauville score, is no longer widely used. In this series, a greater percentage of ≥70-year-old patients had HB < 10.5 and a low lymphocyte count, compared to younger subjects. However, the difference was relevant only for the latter. Neither anemia nor lymphocytes < 600 mcL showed an adverse prognostic impact in univariate analyses.

In about 50% of relapsed/refractory (RR) patients, second-line therapies were reported (Table 4). Unlike first-line treatment, patients aged >70 years mainly received a palliative regimen, and none were referred for ASCT. Conversely, almost 60% of RR patients aged 60–69 years were referred for autologous transplantation.

Despite the fact that GLOBOCAN reports a worse survival for HL patients from Italy, compared to North-Western countries, the outcomes of this series are not substantially different from those of eHL treated with similar approaches in North America. The three-year PFS and OS rates in our survey were 70% and 82%, compared to 61.8% and 83.7% in the report by Orellana-Noia et al. [4]. Notably, in that study, the authors did not observe worse PFS or OS in patients aged ≥70 compared to the 60–69 subgroup, when treated with curative intent. This finding contrasts with both our results and other reports [1,2].

Overall, our results are strengthened by (1) a short enrollment period, (2) a long observation time, and (3) a relatively large sample of eHL patients. Furthermore, the data were collected from a regional network which includes both academic and non-academic hospitals. Conversely, the major biases are the lack of information about geriatric assessments and the retrospective collection of data.

## 5. Conclusions

These real-life data, despite their limitations, show that patients aged 60–69 treated with curative intent achieve excellent long-term EFS, DFS, CSS, and OS, though they may struggle to complete full-dose treatment. Conversely, those aged ≥70 have worse outcomes even after achieving CR.

Therefore, new regimens incorporating CPI and BV with/without chemotherapy will presumably ameliorate the outcome of eHL treated with curative intent. Indeed, the attenuated chemotherapy described in this survey may be a real-life benchmark (Table 2) in comparison with newer approaches. Finally, based on our results, we believe that in eHL with early-stage and bulky disease, RT will also enhance the outcomes of targeted treatments.

## Figures and Tables

**Figure 1 cancers-17-00765-f001:**
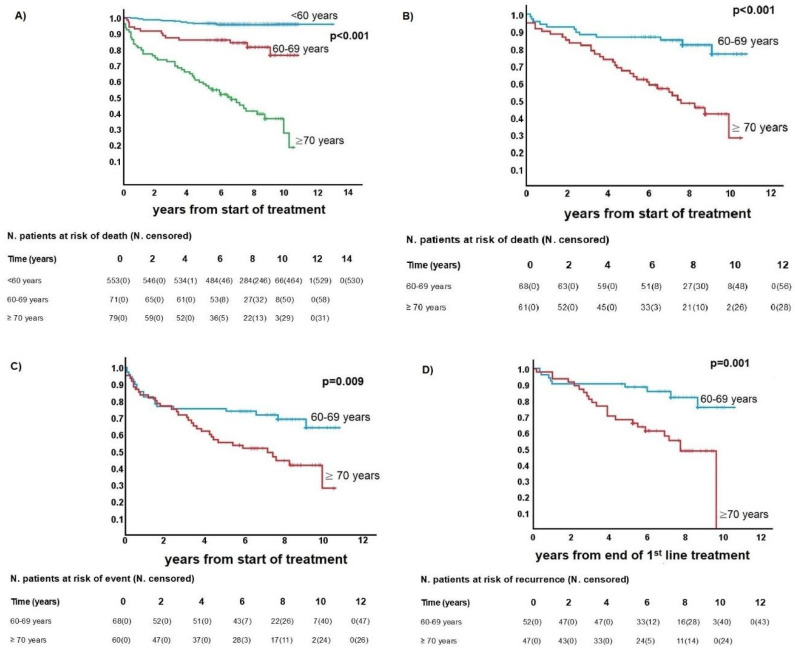
Survival in patients with Hodgkin lymphoma. (**A**) Overall survival of entire population. (**B**) Cancer-specific survival of ≥60-year-old patients treated with curative intent. (**C**) Event-free survival of ≥60-year-old patients treated with curative intent. (**D**) Disease-free survival of ≥60-year-old patients treated with curative intent and complete remission (Kaplan–Meier values 1–5, in Appendix A).

**Figure 2 cancers-17-00765-f002:**
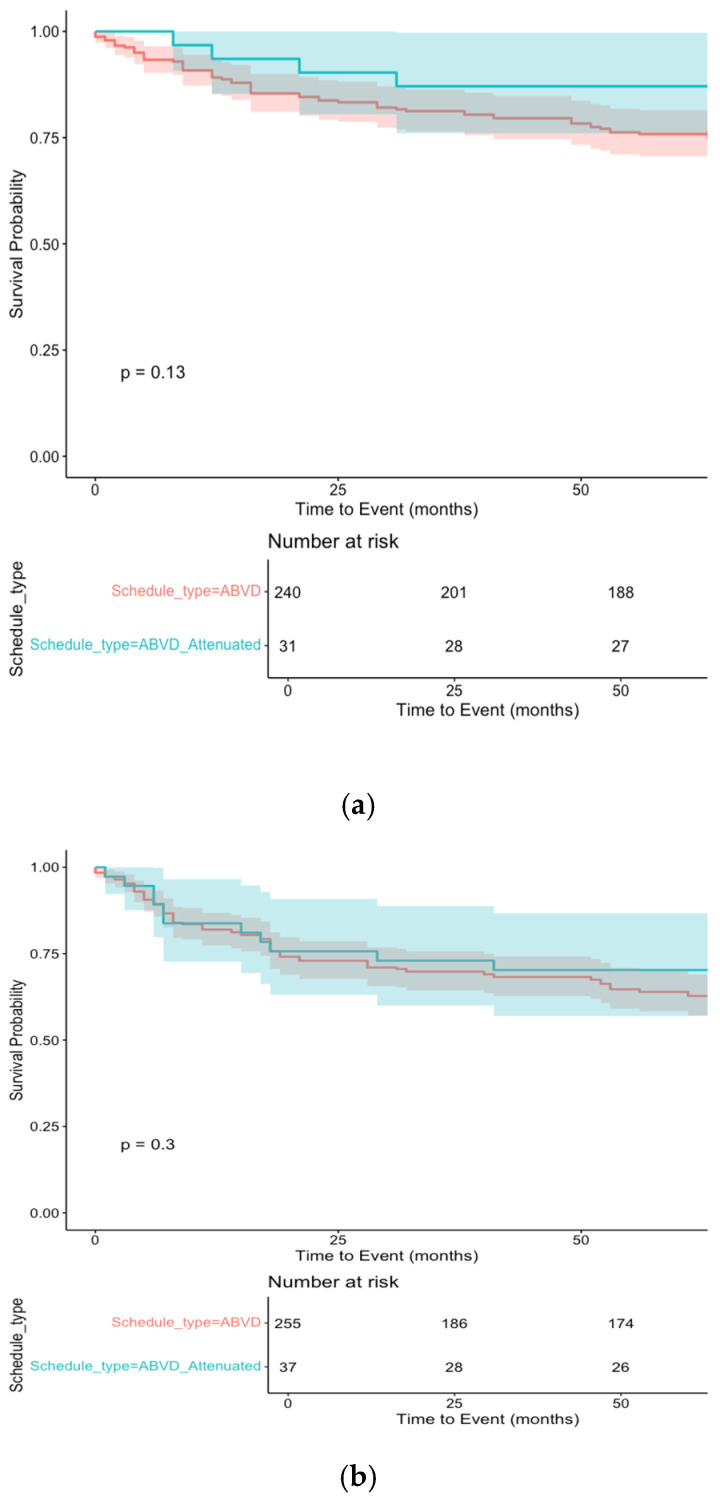
Kaplan–Meier survival curve illustrating time-to-event probability, in first-line chemotherapy schedule type (“ABVD” vs. “Attenuated”) in the eHL synthetic patients. (**a**) Overall survival. (**b**) Event-free survival. Notes: The *x*-axis shows time in months, while the *y*-axis represents cumulative survival probability. Shaded areas denote 95% confidence intervals. The log-rank *p*-value assesses statistical significance between groups.

**Figure 3 cancers-17-00765-f003:**
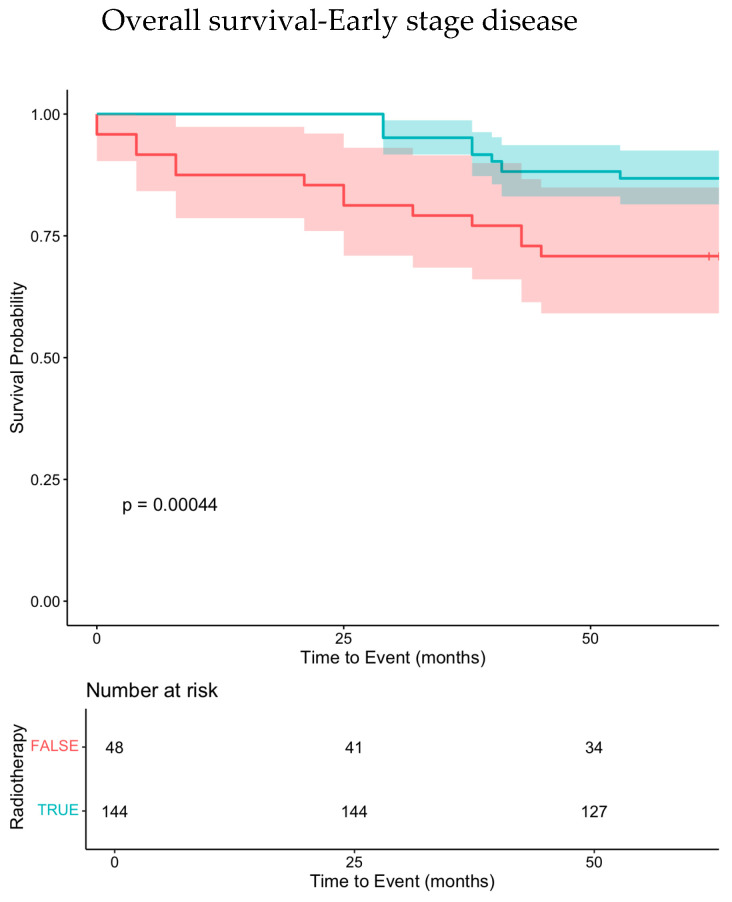
Kaplan–Meier survival curve illustrating time-to-event probability by radiotherapeutic treatment group in the eHL synthetic patients with early-stage disease (stage I–II). Notes: The x-axis shows time in months, while the y-axis represents cumulative survival probability. Shaded areas denote 95% confidence intervals. The log-rank *p*-value assesses statistical significance between groups.

**Table 1 cancers-17-00765-t001:** Demographic and clinical characteristics of Hodgkin lymphoma patients.

Variable	<60 Years Old	60–69 Years Old	≥70 Years Old	*p*-Value
All	556	71	79	
Male sex	48% (266/556)	63% (45/71)	58% (46/79)	0.017
Stage I–II	55% (308/556)	44% (31/71)	30% (24/79)	<0.001
Stage III–IV	45% (248/556)	54% (40/71)	71% (55/79)	<0.001
B symptoms	34% (187/556)	27% (19/71)	29% (23/79)	0.41
ESR > 30	61% (207/340)	66% (27/41)	55% (26/47)	0.60
ESR > 50	40% (137/340)	46% (19/41)	38% (18/47)	0.71
Alb < 3.5 g/dL	18% (81/460)	29% (16/56)	32% (19/60)	0.010
Hb < 10.5 g/dL	16% (75/467)	19% (11/57)	26% (16/61)	0.13
WBCs > 15,000/mmc	13% (62/469)	7% (4/56)	7% (4/61)	0.16
Low lymphocyte count *	12% (58/466)	12% (7/57)	28% (17/60)	0.003
Bulky disease (>7.5 cm)	29% (123/423)	14% (7/51)	6% (3/54)	<0.001
Early-stage with > 3 lymph nodes	37% (114/308)	29% (9/31)	21% (5/24)	0.21
1st-line curative therapy	99% (540/543)	96% (67/70)	76% (52/68)	<0.001
Radiotherapy (1st line)				
All	46% (255/556)	41% (29/71)	25% (20/79)	0.002
Stage I–II	63% (195/308)	68% (21/31)	67% (16/24)	0.85
Stage III–IV	24% (60/248)	20% (8/40)	7% (4/55)	0.020
First-line response after chemotherapy				
CR	79% (406/518)	79% (50/63)	84% (41/49)	0.30
PR	9% (49/518)	15% (9/63)	4% (2/49)	
NR/P	12% (63/518)	6% (4/63)	12% (6/49)	

Alb: albumin; CR: complete response; ESR: erythrocyte sedimentation rate; Hb: hemoglobin; NR: no response; P: progression; PR: partial response; WBCs: white blood cells. * Lymphocytes count < 600 cells/µL OR <8% of total WBC.

**Table 2 cancers-17-00765-t002:** First-line chemotherapies.

Patient No.	Age Class	Schedule Type % (n)		Schedule	n
				ABVD	46
71	60–69	ABVD-like	81.7% (58)	ABVD/AVD	5
				AVD	5
				MBVD/MVD	1
				BEGEV	1
		Attenuated	14.1% (10)	ABVD-like/OPP or COPP	10
		Palliative	4.2% (3)	Bendamustine	1
				COPP	2
				ABVD	39
75	≥70	ABVD-like	69.3% (52)	AVD	3
				MBVD	9
				MVD	1
		Attenuated	10.7% (8)	AVD/OPP	8
		Palliative	20% (15)	Bendamustine	6
				CHLVPP	1
				COPP	4
				OPP	1
				PROVECIP	1
				Vinblastine	2

Abbreviations: ABVD: Adriamycin, Bleomycin, Vinblastine, and Dacarbazine; AVD: Adriamycin, Vinblastine, and Dacarbazine; BEGEV: Bendamustine, Gemcitabine, and Vinorelbine; CHLVVP: Chlorambucil, Vinblastine, Procarbazine, and Prednisone; COPP: Cyclophosphamide, Oncovin, Procarbazine, and Prednisone; MBVD: Myocet, Bleomycin, Vinblastine, and Dacarbazine; MVD: Myocet, Vinblastine, and Dacarbazine; OPP: Oncovin, Procarbazine, and Prednisone; PROVECIP: Procarbazine, Vinblastine, Cyclophosphamide, and Prednisone.

**Table 3 cancers-17-00765-t003:** Second-line treatments.

Patient No.	Age	Second-Line Therapy	% (n)
		BEGEV	13.2 (2)
15	60–69	Brentuximab-Bendamustine	6.7 (1)
		Bendamustine	6.7 (1)
		COPP	6.7 (1)
		CTX	6.7 (1)
		Bendamustine(3) or IGEV(2) or BEGEV(4) + ASCT	60 (9)
17	≥70	Bendamustine	29.4 (5)
		Brentuximab	11.8 (2)
		Chlorambucil	5.9 (1)
		COPP; OPP; PROVECIP; VCR-CTX	23.5 (4)
		DHAOX; IGEV	11.8 (2)
		GEM-VNR	17.6 (3)

Abbreviations: ASCT: Autologous stem cell transplantation; BEGEV: Bendamustine, Gemcitabine, and Vinorelbine; COPP: Cyclophosphamide, Oncovin, Procarbazine, and Prednisone; CTX: Cyclophosphamide; DHAOX: Dexamethasone, Cyrarabine, and Oxaliplatin; GEM: Gemcitabine; IGEV: Ifosfamide, Gemcitabine, and Vinorelbine; OPP: Oncovin, Procarbazine, and Prednisone; PROVECIP: Procarbazine, Vinblastine, Cyclophosphamide, and Prednisone; VCR: Vincristine; VNR: Vinorelbine.

**Table 4 cancers-17-00765-t004:** Univariate and multivariate Cox regression analysis of event-free, overall, and cancer-specific survival in patients > 60 years of age treated with curative intent.

	Event-Free Survival			Overall Survival			Cancer-Specific Survival		
	Univariate	Multivariate		Univariate	Multivariate		Univariate	Multivariate	
	*p*-Value	HR (95% CI)	*p*-Value	*p*-Value	HR (95% CI)	*p*-Value	*p*-Value	HR (95% CI)	*p*-Value
**Age** **(<70 vs. ≥70) years**	**0.008**	1.9 (1.1–3.4)	**0.017**	**<0.001**	3.6 (1.8–7.1)	**<0.001**	**<0.001**	4.9 (2.2–10.8)	**<0.001**
**Gender** **(M vs. F)**	0.96	-	-	0.50	-	-	0.19	-	-
**Stage** **(I–II vs. III–IV)**	**0.008**	1.1 (0.6–2.1)	0.76	**0.028**	0.9 (0.4–1.8)	0.72	**0.046**	0.7 (0.3–1.6)	0.42
**B symptoms** **(no vs. yes)**	0.70	-	-	0.39	-	-	0.53	-	-
**Bulky disease** **(no vs. yes)**	0.95	-	-	0.86	-	-	0.49	-	-
**Hemoglobin** **(<10.5 vs. ≥10.5) g/dL**	0.23	-	-	0.10	-	-	0.21	-	-
**WBCs** **(<15,000 vs. ≥15,000) μL^−1^**	0.065	-	-	0.77	-	-	0.52	-	-
**Lymphocytes** **(≤600 vs. >600) μL^−1^**	0.17	-	-	0.17	-	-	0.12	-	-
**Radiotherapy** **(no vs. yes)**	**0.001**	3.3 (1.6–7.1)	**0.002**	**<0.001**	4.9 (1.9–12.5)	**0.001**	**0.001**	6.9 (2.2–21.2)	**0.001**
**Complete remission** **(no vs. yes)**	**<0.001**	0.2 (0.1–0.3)	**<0.001**	**<0.001**	0.2 (0.1–0.4)	**<0.001**	**<0.001**	0.2 (0.1–0.4)	**<0.001**
**PET 2** **(no vs. yes)**	0.41	-	-	0.90	-	-	0.81	-	-

Abbreviations: CI: confidence interval; F: female; HR: hazard ratio; M: male; PET: positron emission tomography WBCs: white blood cells. Bold numbers are significant values.

## Data Availability

The dataset is available upon request (chrisscox@gmail.com; stefan.hohaus@unicatt.it).

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
