# Peer review of "The Impact of Radiotherapy and Attenuated Chemotherapy Regimens in Older Patients with Classic Hodgkin Lymphoma: A Real-Life Study from the ReLLi Network"

_cancers, 2025, doi:10.3390/cancers17050765_

Round 1
Reviewer 1 Report
Comments and Suggestions for Authors
In the submitted manuscript the authors describe and discuss the results of a multicenter retrospective study that focused on the clinical impact of radiotherapy and attenuated chemotherapy regimens in elderly patients with classical Hodgkin lymphoma (eHL).
The Introduction provides the background for this study, being highlighted the clinical particularities of eHL and the therapeutic challenges encountered in the current practice. The objectives of the study are clearly presented. The research was properly designed, being compared the disease characteristics and treatment outcome of the eHL with those of younger HL patients diagnosed over the same period of six years. The assessment of clinico-biological parameters and of treatment endpoints is well performed and the statistical results are adequately displayed. The generation of synthetic data is also very relevant for the study.
There are a few minor issues to be addressed:
1. Although the ECOG Performance Status was mentioned as a variable collected from the medical records it was not included in the assessment.
2. In Table 1 the low lymphocyte count, and not anemia, is significantly more often encountered in eHL. Please, comment on it accordingly.
3. At Discussion section, please clarify the statements: „These data, suggest RT, in 70+ subjects, with limited stage or bulky disease, may be beneficial even when there are no residual lesions and PET is negative. Indeed, the very low rate of RT in 70+ years patients in advanced stage, is presumably due to the presence of widespread disease”.
Author Response
In the submitted manuscript the authors describe and discuss the results of a multicenter retrospective study that focused on the clinical impact of radiotherapy and attenuated chemotherapy regimens in elderly patients with classical Hodgkin lymphoma (eHL).
The Introduction provides the background for this study, being highlighted the clinical particularities of eHL and the therapeutic challenges encountered in the current practice. The objectives of the study are clearly presented. The research was properly designed, being compared the disease characteristics and treatment outcome of the eHL with those of younger HL patients diagnosed over the same period of six years. The assessment of clinico-biological parameters and of treatment endpoints is well performed and the statistical results are adequately displayed. The generation of synthetic data is also very relevant for the study.
- Although the ECOG Performance Status was mentioned as a variable collected from the medical records it was not included in the assessment.
Response: we apologize our mistake, unfortunately we could not collect the ECOG. Therefore we omitted the term in the manuscript.
- In Table 1 the low lymphocyte count, and not anemia, is significantly more often encountered in eHL. Please, comment on it accordingly.
Response: we are grateful to the Reviewer for his comment, which has contributed to implement the discussion, as follows (lines 627-633):“Few data are available regarding laboratory parameters in older eHL and their impact on prognosis [ 31]. Both anemia and low lymphocytes count are included in the IPS score [32], which after the advent of interim-PET Deauville score, is not anymore widely used. In this series, a greater percentage of ≥70 had HB<10.5 and a low lymphocyte count, compared to younger subjects. However, the difference was relevant only for the latter. Neither anemia nor lymphocytes<600mcL showed an adverse prognostic impact in univariate analyses”.
- At Discussion section, please clarify the statements: „These data, suggest RT, in 70+ subjects, with limited stage or bulky disease, may be beneficial even when there are no residual lesions and PET is negative. Indeed, the very low rate of RT in 70+ years patients in advanced stage, is presumably due to the presence of widespread disease”.
Response: The paragraph was re-phrased as follow: “These data, suggest RT, in subjects aged ≥70, with limited stage, may be beneficial even when there are no residual lesions and interim PET is negative. Only 7% of ≥70 patients in advanced stage, received RT, this is presumably due to the very low incidence of bulky disease in eHL compared to younger patients” (lines 462-465)
Reviewer 2 Report
Comments and Suggestions for Authors
The present manuscripts reports real-world data from elderly patients with classic Hodgkin lymphoma treated in Italy. Although the analysis includes a significant number of patients so that conclusions can be drawn, there are different points that require revision or clarification:
1) rather use the term classic Hodgkin lymphoma than classical Hodgkin lymphoma
2) Abstract: please mention the number of patients in the different age groups already in the abstract
3) Introduction: patients were treated between 2013 and 2018. This is an advantage given the mature follow-up but also a disadvantage as BV and CPI were not yet available in this period . This should be mentioned already in the introduction
4) Methods: please clarify the definition of "reduced-dose anthracyclin"
5) Discussion: The fact that RT improves disease control in early-stage disease even in patients with a CR after chemotherapy has been demonstrated by three randomized studies (H10, HD16, RAPID). This should be mentioned and discussed (based on the most recent analyses of these trials)
6) Discussion: concerning bleomycin, please discuss the RATHL trial and the findings from German Hodgkin Study Group trials with regard to the use of this drug in older patients
7) Discussion: please discuss the results of the subgroup analysis on older patients from the SWOG S1826 trial indicating a clear advantage of Nivo-AVD over BV-AVD
8) General: The discussion rather summarizes the results instead of putting them into context and thus requires substantial revision
Comments on the Quality of English Language
n/a
Author Response
The present manuscripts reports real-world data from elderly patients with classic Hodgkin lymphoma treated in Italy. Although the analysis includes a significant number of patients so that conclusions can be drawn, there are different points that require revision or clarification:
- rather use the term classic Hodgkin lymphoma than classical Hodgkin lymphoma
Response: This issue was addressed all over the text
- Abstract: please mention the number of patients in the different age groups already in the abstract
Response: These figures are now in the revised abstract
- Introduction: patients were treated between 2013 and 2018. This is an advantage given the mature follow-up but also a disadvantage as BV and CPI were not yet available in this period . This should be mentioned already in the introduction
Response: This is now mentioned in the introduction section (lines 112-113): “when Brentuximab-vedotin (BV) and checkpoint inhibitors (CPI) were not yet available”
- Methods: please clarify the definition of "reduced-dose anthracyclin"
Response: This was clarified in the methods section (lines 247-250): "ABVD-like regimens," (i.e., full-dose anthracycline-based chemotherapy. "ABVD-like regimens," (i.e., full-dose anthracycline-based chemotherapy. "Attenuated regimens" (schedules that alternate cycles of full-dose anthracycline with cycles that do not include anthracyclines, such as OPP and COPP), “Palliative regimens” (i.e. not curative),
- Discussion: The fact that RT improves disease control in early-stage disease even in patients with a CR after chemotherapy has been demonstrated by three randomized studies (H10, HD16, RAPID). This should be mentioned and discussed (based on the most recent analyses of these trials)
Response: we thank the Reviewer for his comment, and added discussion of the most recent papers on the three trials mentioned by the reviewer, as follows (lines 531-540): “This further highlights the critical role of radiotherapy in the therapeutic management and outcomes of eHL patients. Three randomized trials (H10 trial of the EORTC, the HD16 trial of the GSHG and the RAPID trial from the UK) demonstrated that omission of radiotherapy in patients with limited stage disease and an early complete metabolic response associates with inferior progression-free survival [17-20], but failed to show an impact on OS. These trials included also a minority of elderly patients up to 70 years in the H10 trial and 75 years in the HD16 and RAPID trial. A detailed analysis on this small subgroup of elderly subjects has not been reported. As far as we know, the impact of radiotherapy on OS has not been analyzed in other real-life series focused on eHL [2, 4].”
6) Discussion: concerning bleomycin, please discuss the RATHL trial and the findings from German Hodgkin Study Group trials with regard to the use of this drug in older patients
Response: we addressed this issue inserting a new paragraph (547-555): “. Johnson et al. [21] demonstrated that, with a PET-oriented approach, it is possible to remove bleomycin (AVD) after two cycles of ABVD without reducing treatment efficacy in terms of both PFS and OS [21]. Moreover, respiratory adverse events were more severe in the ABVD arm than in the AVD arm. The German Hodgkin Study Group (GHSG) also showed that, in localized stages, limiting bleomycin—particularly in elderly patients—to the first two cycles of ABVD was less toxic while maintaining comparable efficacy. Recently, the German group eliminated bleomycin, among other agents, in the new BrECADD scheme [22,23]”
- Discussion: please discuss the results of the subgroup analysis on older patients from the SWOG S1826 trial indicating a clear advantage of Nivo-AVD over BV-AVD
Response: we discussed the results of the trial as follow (562-576): “Currently, the results of several trials, enrolling eHL, and based on new combination regimens have become available [24,25] Indeed, the introduction of new antibodies, particularly in sequential regimens, is expected to improve outcomes in this subset [24]. In fact, recently it was reported that AAVD is less effective and more toxic in older populations [26]. Conversely, Evens' study [27] demonstrates that the sequential use of immunotherapy with chemotherapy enhances results in elderly patients by reducing toxicity and enabling more patients to complete treatment without adverse events leading to discontinuation. Also, the results of the pivotal phase III SWOG S1826 study [28] were published. Although this trial was not specifically designed to address the treatment of elderly patients, a sub-analysis clearly demonstrated that in fit older patients (over 60 years), the N-AVD combination resulted in significantly better outcomes compared to BV-AVD, both in terms of tolerability and efficacy [28]. However, Nivo-AVD, may increase the risk of immune-related cardiotoxicity [28,29]”.
- General: The discussion rather summarizes the results instead of putting them into context and thus requires substantial revision
Response: we believe that the discussion has been significantly improved based on the Reviewers' insightful comments. We appreciate their feedback, which has helped refine our analysis and interpretation. Furthermore, the manuscript has been thoroughly reviewed by a native English-speaking language professional
- Please increase the number of references to at we least 30.
Response: The references were implemented up to 32 citations
Round 2
Reviewer 2 Report
Comments and Suggestions for Authors
The revised version of the present manuscript has addressed most of the points raised by the reviewer. However, there are still few points that require additional revision:
1) Abstract: rather write treated between 2013 and 2018 than only "between 2013 and 2018"
2) Abstract and results: OS at what time? 5-year OS?
3) Abstract and Methods section: the abbreviation CSS has different definitions. Please clarify.
Comments on the Quality of English LanguageThe quality of grammar and spelling can be improved throughout the manuscript
Author Response
We thank the reviewer, for his further observation and suggestions
1) Abstract: rather write treated between 2013 and 2018 than only "between 2013 and 2018"
R: this was addressed in the abstract
2) Abstract and results: OS at what time? 5-year OS?
This specification was made for OS, CSS and EFS, both in the abstract and in the result section
3) Abstract and Methods section: the abbreviation CSS has different definitions. Please clarify.
CSS stands for cancer specific survival, this was clarified in the methods section